# Quantification of stroke volume in a simulated healthy volunteer model of traumatic haemorrhage; a comparison of two non-invasive monitoring devices using error grid analysis alongside traditional measures of agreement

**Sam D. Hutchings**[1,2]*, **Jim Watchorn**[1], **Rory McDonald**[2], **Su Jeffreys**[2], **Mark Bates**[3], **Sarah Watts**[3], **Emrys Kirkman**[3]

1 Department of Inflammation Biology, School of Immunology and Microbial Sciences, King's College London, London, United Kingdom, 2 Academic Department of Military Anaesthesia and Critical Care, Royal Centre for Defence Medicine, Birmingham, United Kingdom, 3 Defence Science and Technology Laboratory, Porton Down, Wiltshire, United Kingdom

* sam.hutchings@nhs.net

## Abstract

### Introduction

Haemorrhage is a leading cause of death following traumatic injury and the early detection of hypovolaemia is critical to effective management. However, accurate assessment of circulating blood volume is challenging when using traditional vital signs such as blood pressure. We conducted a study to compare the stroke volume (SV) recorded using two devices, trans-thoracic electrical bioimpedance (TEB) and supra-sternal Doppler (SSD), against a reference standard using trans- thoracic echocardiography (TTE).

### Methods

A lower body negative pressure (LBNP) model was used to simulate hypovolaemia and in half of the study sessions lower limb tourniquets were applied as these are common in military practice and can potentially affect some haemodynamic monitoring systems. In order to provide a clinically relevant comparison we constructed an error grid alongside more traditional measures of agreement.

### Results

21 healthy volunteers aged 18–40 were enrolled and underwent 2 sessions of LBNP, with and without lower limb tourniquets. With respect to absolute SV values Bland Altman analysis showed significant bias in both non-tourniquet and tourniquet strands for TEB (-42.5 / -49.6 ml), rendering further analysis impossible. For SSD bias was minimal but percentage error was unacceptably high (35% / 48%). Degree of agreement for dynamic change in SV,

**Data Availability Statement:** All relevant data are within the manuscript and its Supporting Information files.

**Funding:** Funding for the study was provided by the Defence Science and Technology Laboratory, part of the UK Ministry of Defence. The funders had no role in study design, data collection and analysis, decision to publish, or preparation of the manuscript.

**Competing interests:** The authors have declared that no competing interests exist.

assessed using 4 quadrant plots showed a seemingly acceptable concordance rate for both TEB (86% / 93%) and SSD (90% / 91%). However, when results were plotted on an error grid, constructed based on expert clinical opinion, a significant minority of measurement errors were identified that had potential to lead to moderate or severe patient harm.

## Conclusion

Thoracic bioimpedance and suprasternal Doppler both demonstrated measurement errors that had the potential to lead to clinical harm and caution should be applied in interpreting the results in the detection of early hypovolaemia following traumatic injury.

## Introduction

Haemorrhage remains the leading cause of preventable death following traumatic injury [1]. Although the diagnosis of severe haemorrhage is often easy in retrospect, it can be challenging to make contemporaneously, especially as patients typically present to medical providers in austere settings far removed from the monitoring facilities available in the operating theatre or intensive care unit.

Detection of haemorrhage induced hypovolaemia is usually based on assessment of so called traditional vital signs, particularly blood pressure. However, the use of arterial blood pressure to estimate the degree of blood loss is problematic and potentially inaccurate [2–4] and there is a poor association between recorded blood pressure and tissue perfusion parameters [5]. A monitoring system that could detect changes in flow and volume based parameters may allow better detection of early hypovolaemia in patients at risk of blood loss.

Obtaining precise effect and response data as it relates to the physiology of haemorrhagic shock is complicated by the uncontrolled nature of the clinical environment, an alternative approach is to study healthy volunteers using lower body negative pressure (LBNP), a technique which produces central hypovolaemia, and therefore simulates the early stages of haemorrhage. Progressive application of LBNP is a well-established technique used to simulate haemorrhage as well as a range of other conditions including syncope associated with orthostasis [6, 7]. A subject is asked to lie supine with the lower portion of the body (from the waist down) sealed in a chamber. The pressure within the chamber is reduced to sub-atmospheric levels (application of LBNP), which results in blood pooling in the veins of the legs and pelvis. Since the capacity of the pelvic veins is much greater than those of the legs, it is the pelvic veins that make the greatest contribution to this blood pooling. As the LBNP is gradually augmented (made more negative with respect to atmospheric), progressively greater amounts of blood are trapped in the lower body and there is a corresponding decrease in venous return to the heart, simulating the effects of haemorrhage. Using this technique it is possible to elicit both the initial compensatory response to haemorrhage (tachycardia while blood pressure is maintained) and the later depressor phase (reflex bradycardia and hypotension) leading to pre-syncope and then syncope. This pattern closely mirrors the cardiovascular changes seen with actual haemorrhage [8–11]. We used an LBNP model in order to compare the performance of two non-invasive monitoring devices that utilised thoracic electrical bioimpedance and supra-sternal Doppler to measure stroke volume. Trans-thoracic echocardiography derived stroke volume was used as the standard or reference measurement. As the application of lower limb tourniquets to prevent catastrophic haemorrhage is not uncommon following major trauma, and may produce an effect on arterial waveforms and hence influence the output of some monitoring devices, we applied lower limb tourniquets during half of the study sessions.

In order to assess the clinical relevance, as well as the mathematical agreement between the measurements produced we utilised an error grid analysis approach alongside more traditional measures of agreement.

## Materials and methods

### Ethical approval

The study received ethical approval from the Ministry of Defence Research Ethics Committee (538/MODREC/14). Subjects provided written informed consent prior to study enrolment.

### Study cohort

21 healthy volunteers aged between 18 and 40. The sample size was based on 75% power to detect a 10% difference in SV between candidate devices, with an alpha of 0.05.

### Recruitment screening and exclusion criteria

Volunteers filled out a medical questionnaire and were interviewed by a study investigator prior to enrolment. They also underwent 12-lead ECG and cardiac echocardiography examination. Volunteers were not eligible for enrolment if they had a history, or any symptoms or signs, of cardiovascular disease. Individuals judged to be at risk of thromboembolic complications (family history, recent lower limb or pelvic trauma or current usage of oral contraceptive medication) were excluded.

### Lower body negative pressure protocol

Volunteers lay supine with the lower portion of the body, from the waist down, sealed in a chamber. The pressure within the chamber was reduced to sub-atmospheric levels using a vacuum generator. A continuous, real time, measure of reconstructed brachial artery blood pressure (rBAP) was made using a Finometer® PRO (Finapres Medical Systems BV, Netherlands) via a cuff placed on the right middle finger. The rBAP was continuously normalised to heart level (reference zero) using a hydrostatic height sensor affixed to the volunteer's chest at the mid-axial line, and to the finger cuff. Cardiac stroke volume ($SV_{Fin}$) was derived continuously using the built-in Modelflow® technology, which has been shown to be reliable in reporting trend changes in cardiac output [12], and hence stroke volume. The cardiovascular data (rBAP and $SV_{Fin}$) were exported continuously from the Finometer® PRO via a digital to analogue converter and recorded on the computerized data acquisition system (MacLab 8/s). Heart rate was derived from the rBAP trace using the LabChart®7 PRO software.

Each subject rested supine in the LBNP chamber for 20 min after instrumentation to allow a steady state to be reached. Thereafter two baseline readings of all cardiovascular variables were made at 5 min intervals. Average data from the final 3 minutes of the second baseline period was used to calculate each subject's baseline $SV_{Fin}$ for that session. Negative pressure was then applied within the chamber to trap blood in the lower body causing central hypovolaemia. By sequentially applying steps of negative pressure, a progressive haemorrhage was simulated. The initial two steps were to an absolute pressure level of 10 and 20 mmHg below atmospheric, and thereafter to attain a target $SV_{Fin}$ of 80%, 65–70% and 55–64% of the baseline recorded in the subject. LBNP was then reduced in one or two steps to a final level of 0 mmHg (no suction). Each step lasted approximately 5 min. Stroke volume measurements from the three monitoring devices (trans-thoracic echocardiography, thoracic bioreactance and suprasternal Doppler) were recorded approximately 2 minutes into each step.

Sessions involving tourniquet application were conducted in an identical manner to those without tourniquet, except that bilateral tourniquets (Conical Leg Tourniquets, SCT2x2, Braun & Co Ltd, UK) were inflated around the upper thighs to a pressure of 100 mmHg above the subject's systolic arterial pressure immediately after the second baseline measurement. Two minutes after tourniquet inflation all cardiovascular measurements were made before commencing the LBNP protocol described above. The tourniquets remained inflated without further adjustment of pressure until the end of the LBNP protocol.

## Trans Thoracic Echocardiography (TTE)

Serial focussed trans thoracic echocardiography studies were performed (Sparq, S41 transducer, Philipps UK) by a single operator, accredited with the British Society of Echocardiography. Measurement of the left ventricular outflow tract (LVOT) diameter was obtained using a parasternal long axis window prior to the commencement of LBNP. Serial measurements of $SV_{TTE}$ were obtained by applying continuous wave Doppler to the LVOT in a 5 chamber view and calculating the velocity time integer (VTi) of the resultant signal.

## Supra Sternal Doppler (SSD)

Serial measurements were made of the blood flow in the ascending aorta / aortic arch using supra sternal Doppler (Ultra Sonic Cardiac Output Monitor, USCOM 1A, USCOM, Australia). A single trained user conducted all measurements. The software automatically produces a value for $SV_{SSD}$ based on subject demographics (age, weight, height and gender) and a calculation of VTi. Supra-sternal Doppler measurements were taken immediately following TTE.

## Thoracic electrical bio-impedance (TEB)

Serial stroke volume measures were taken using a system that utilises trans-thoracic bio-impedance (NICOM®, Cheetah Medical) This technique relies on the phase shift of electric current between four surface electrodes placed on the thorax. A continuous reading for $SV_{TEB}$ was produced and recorded synchronously with $SV_{TTE}$ measurements.

## Data analysis and statistical methods

Distribution of data was assessed using D'Agostino & Pearson omnibus normality test. Parametric continuous data is presented as mean ± standard deviation (SD). Statistical analysis was performed using Graphpad Prism v. 6 and NCSS v. 11. p values of $<0.05$ was taken as statistically significant.

Changes in cardiovascular parameters were analysed by linear mixed model ANOVA with repeated measures over time, using baseline values as a covariate. Data are presented as mean ±SEM.

Comparison between absolute values of reference stroke volume ($SV_{TTE}$) and those recorded by supra-sternal Doppler ($SV_{SSD}$) and thoracic electrical bioimpedance ($SV_{TEB}$) was performed using Bland-Altman analysis [13] adapted for repeated measurements of a range of stroke volumes in the series of volunteers; the percentage error was calculated by dividing the limit of agreement (1.96 SD) by the mean value for $SV_{TTE}$.

Changes in stroke volume from baseline values recorded by trans-thoracic echocardiography ($\Delta SV_{TTE}$), supra-sternal Doppler ($\Delta SV_{SSD}$) and thoracic bioimpedance ($\Delta SV_{TEB}$) was compared using four quadrant plots with a 15% central exclusion zone applied [14, 15]. Correlation between $\Delta SV_{TTE}$ and $\Delta SV_{TBI}$ / $\Delta SV_{SSD}$ was assessed using Pearson's correlation coefficient.

In order to assess the clinical significance of differences in SV measurements recorded by the subject devices we utilised an error grid analysis approach [16]. Firstly, we distributed a questionnaire to intensive care and anaesthesia specialists at our institution and in the United Kingdom Defence Medical Services (S1 File). We asked them to envisage a clinical scenario in which they were managing a previously fit patient at risk from hypovolaemia following traumatic injury and to consider that they were using a device to monitor stroke volume and that the therapeutic intervention at their disposal was administration of blood products to treat hypovolaemia. We firstly asked them to categorise the percentage fall in SV which they would regard as requiring i) no current action ii) action indicated iii) action essential.

We then asked the clinicians to consider, in the light of these responses, the potential for harm resulting from a divergence in SV change between the actual value and the value recorded by a measurement device. We asked respondents to categorise this potential for harm as: None, Mild, Moderate or Severe. Based on these responses we constructed an error grid using the following methodology:

i. A numerical weighting factor was applied to the potential risk of harm as follows: None 0, Mild 2, Moderate 5, Severe 10. This weighting factor is necessarily subjective.

ii. A table was created from the combined questionnaire returns which compared actual SV reduction against measured SV reduction (in the range 0 to -60%). Average risk weighted numerical scores, based on the questionnaire responses, were entered into each cell of this table. This produced a numeric range from 0 (indicating no respondent thought there was no potential for harm) to 150 (indicating that all respondents felt that there was a severe risk of harm). (Supplementary material–error grid returns).

iii. An MS Excel spreadsheet was created detailing all potential values for SV reduction recorded by device against actual SV reduction in a range from 0 to 60% and values were then converted into a percentage and a Red–Yellow–Green colour characteristic applied to all cells such that the risk of harm was illustrated graphically from Green (0) to Red (150). Supplementary material–error grid spreadsheet.

iv. An image of the colour coded spreadsheet was exported and polygons drawn over areas of similar colour in order to create 4 zones representing no risk (green), mild risk (yellow), moderate risk (orange) and severe risk (red). (Supplementary material—error grid polygons).

v. The image was imported into a graphing program (Desmos, https://www.desmos.com). Values for TTE derived SV change were plotted on the x axis and values derived from the assessed measurement devices were plotted on the y axis.

vi. The number of recorded values falling into each area of risk was recorded for each measurement device.

## Results

### Participant demographics

Healthy volunteers were predominantly male 13/21 (62%), aged 27±5 years and had a body mass index of 23±3 kg/m$^2$.

### Cardiovascular response to LBNP

The overall cardiovascular response to LBNP are shown in Fig 1, and the corresponding statistical output is shown in Table 1. There was a significant reduction in stroke volume, rise in

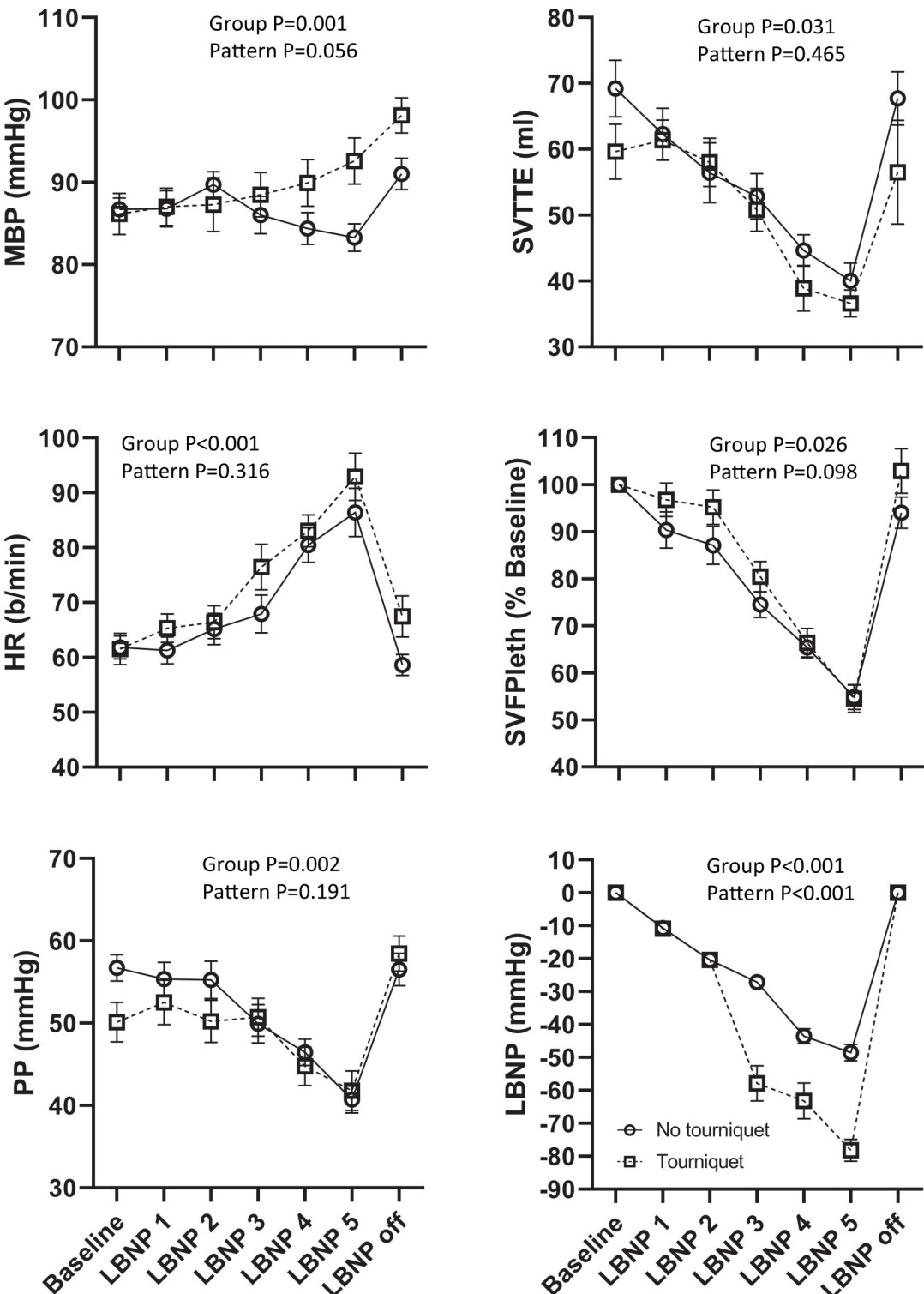

**Fig 1. Cardiovascular response to progressive LBNP.** Effects of progressive LBNP (-10, -20 mmHg, then titrated to target $SV_{FPleth}$, stroke volume measured by finger Plethysmography, of 80, 65–70 and 55–64% baseline $SV_{FPleth}$, LBNP1-5 respectively) on mean arterial blood pressure (MBP), heart rate (HR), arterial pulse pressure (PP), stroke volume measured by trans-thoracic echo ($SV_{TTE}$). Data presented as mean±SEM. p values indicated on each panel show probability of differences between groups, and differences in the pattern of changes, being due to chance (ANOVA). Changes over time were statistically significant (p <0.001) for each variable.

**Table 1. Summary repeated measures ANOVA output for the cardiovascular parameters and LBNP levels shown in Fig 1.**

| Parameter | Effect type | Term | F Df(Num, Den) | F-Value | P-Value |
|---|---|---|---|---|---|
| $SV_{FPleth}$ (% baseline) | Main | LBNP sequence | 6, 205 | 265.27 | <0.001 |
| | Main | Strand | 1, 215 | 5.03 | 0.026 |
| | Interaction | LBNP*Strand | 6, 205 | 1.81 | 0.098 |
| $SV_{TTE}$ | Main | LBNP sequence | 6, 195 | 122.96 | <0.001 |
| | Main | Strand | 1, 212 | 4.72 | 0.031 |
| | Interaction | LBNP*Strand | 6, 195 | 0.94 | 0.465 |
| HR | Main | LBNP sequence | 6, 204 | 55.13 | <0.001 |
| | Main | Strand | 1, 208 | 15.43 | <0.001 |
| | Interaction | LBNP*Strand | 6, 204 | 1.18 | 0.316 |
| PP | Main | LBNP sequence | 6, 200 | 47.09 | <0.001 |
| | Main | Strand | 1, 217 | 9.71 | 0.002 |
| | Interaction | LBNP*Strand | 6, 200 | 1.47 | 0.191 |
| MAP | Main | LBNP sequence | 6, 203 | 5.32 | <0.001 |
| | Main | Strand | 1, 210 | 7.32 | 0.007 |
| | Interaction | LBNP*Strand | 6, 202 | 2.09 | 0.056 |
| LBNP | Main | LBNP sequence | 6, 204 | 284.32 | <0.001 |
| | Main | Strand | 1, 210 | 99.88 | <0.001 |
| | Interaction | LBNP*Strand | 6, 204 | 21.46 | <0.001 |

Abbreviations for the parameters are as listed in Fig 1. The main effects of LBNP sequence (Time) and Strand (Tourniquet / No tourniquet), and the interaction of the two main effects (LBNP*Time) are shown. F Df(Num, Den), F degrees of freedom (Numerator, Denominator). $SV_{FPleth}$ stroke volume finger plehysmography, $SV_{TTE}$ stroke volume echocardiography, HR heart rate, PP pulse pressure, MAP mean arterial pressure, LBNP lower body negative pressure.

heart rate and fall in pulse pressure as LBNP was progressively increased. The overall pattern of response was similar in both the tourniquet and non- tourniquet strands with the exception of blood pressure, which fell slightly in the non-tourniquet strand and rose in the tourniquet strand.

$SV_{TTE}$ generally correlated well with $SV_{FIN}$ (Pearson's $R^2 > 0.7$ in 100% of subjects in the non-tourniquet stream and 88% of subjects in the tourniquet stream).

## Comparison of absolute values of $SV_{TTE}$ and $SV_{TEB}$ / $SV_{SSD}$

Bland Altman plots comparing $SV_{TTE}$ and $SV_{TEB}$ are shown in Fig 2. 144 paired measurements were made in the non-tourniquet strand and 138 in the tourniquet strand. Due to the fact that bias obviously increased with increasing mean SV, limits of agreement are not shown for this comparison; percentage error between $SV_{TTE}$ and $SV_{TEB}$ was not calculated for the same reason.

Bland Altman plots comparing $SV_{TTE}$ and $SV_{SSD}$ are shown in Fig 3. 152 paired measurements were made in the non-tourniquet strand with a bias of 0.1ml and limits of agreement between 36.7 ml and -36.5ml. 139 paired measurements were made in the tourniquet strand with a bias of -4.6ml and limits of agreement between 22.4 ml and -31.6 ml. Percentage error between $SV_{TTE}$ and $SV_{SSD}$ was 35% in the non-tourniquet strand and 48% in the tourniquet strand.

## Comparison of change from baseline values for $SV_{TTE}$ and $SV_{TEB}$ / $SV_{SSD}$

Four quadrant plots comparing changes in SV from baseline for TTE ($\Delta SV_{TTE}$) and SSD ($\Delta SV_{SSD}$) are shown in Fig 4. In the non-tourniquet strand 119 paired measurements were made, $\Delta SV_{TTE}$ and $\Delta SV_{SSD}$ were positively correlated (r = 0.65, p<0.0001) with a 91%

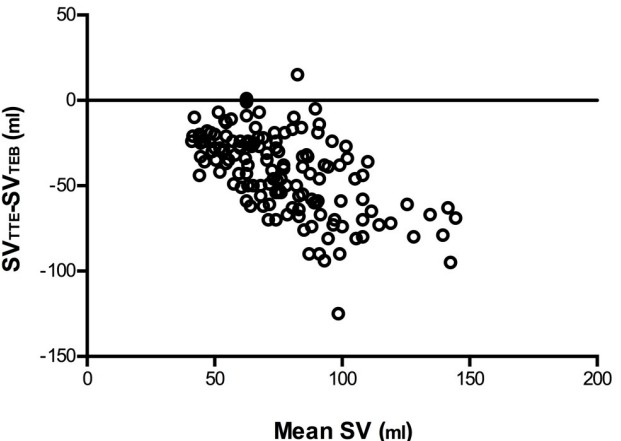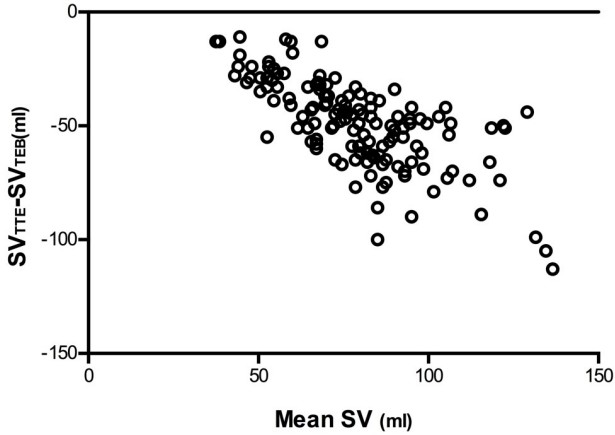

### No Tourniquet stream

### Tourniquet stream

**Fig 2. Bland-Altman plot comparing stroke volume (SV) values recorded using thoracic bioimpedence (TBI) and trans-thoracic echocardiography (TTE).** Values indicate bias±SD and 95% limits of agreement.

concordance rate. In the tourniquet strand 113 paired measurements were made, $\Delta SV_{TTE}$ and $\Delta SV_{SSD}$ were positively correlated (r = 0.75, p<0.0001) with a concordance rate of 90%.

Four quadrant plots comparing changes in SV from baseline for TTE ($\Delta SV_{TTE}$) and TEB ($\Delta SV_{TEB}$) are shown in Fig 5. In the non-tourniquet strand 121 paired measurements were made, $\Delta SV_{TTE}$ and $\Delta SV_{TEB}$ were positively correlated (r = 0.41, p<0.0001) with a 86% concordance rate. In the tourniquet strand 112 paired measurements were made, $\Delta SV_{TTE}$ and $\Delta SV_{TEB}$ were positively correlated (r = 0.78, p<0.0001) with a 93% concordance rate.

### Error grid analysis

Responses from 15 specialists were used to construct an error grid. Responses are provided in the (S2 File). The resultant spreadsheet used to create the error grid and the polygon overlay template created from the spreadsheet are provided in the (S3 File and S1 Fig).

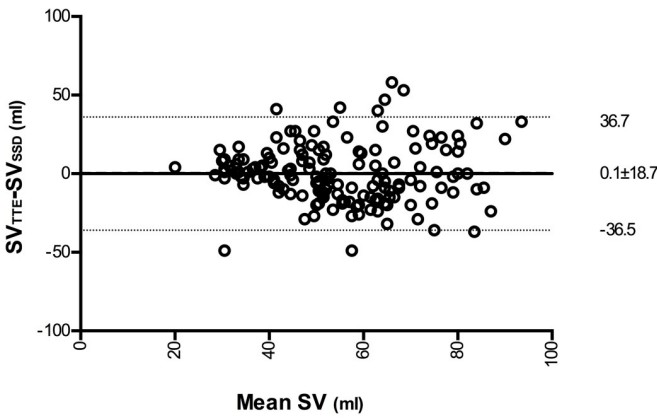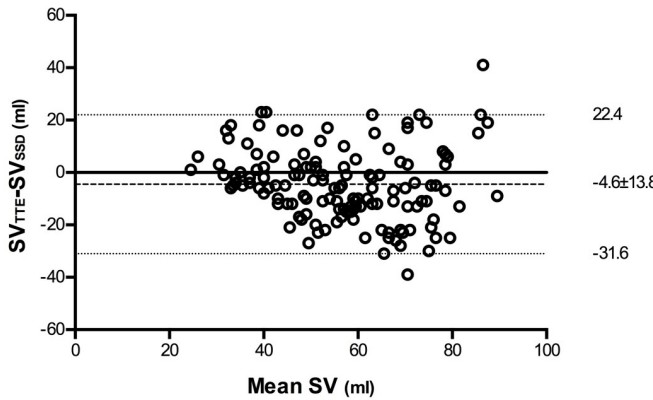

### No Tourniquet stream

### Tourniquet stream

**Fig 3. Bland-Altman plot comparing stroke volume (SV) values recorded using supra-sternal Doppler (SSD) and trans-thoracic echocardiography (TTE).** Values indicate bias±SD and 95% limits of agreement.

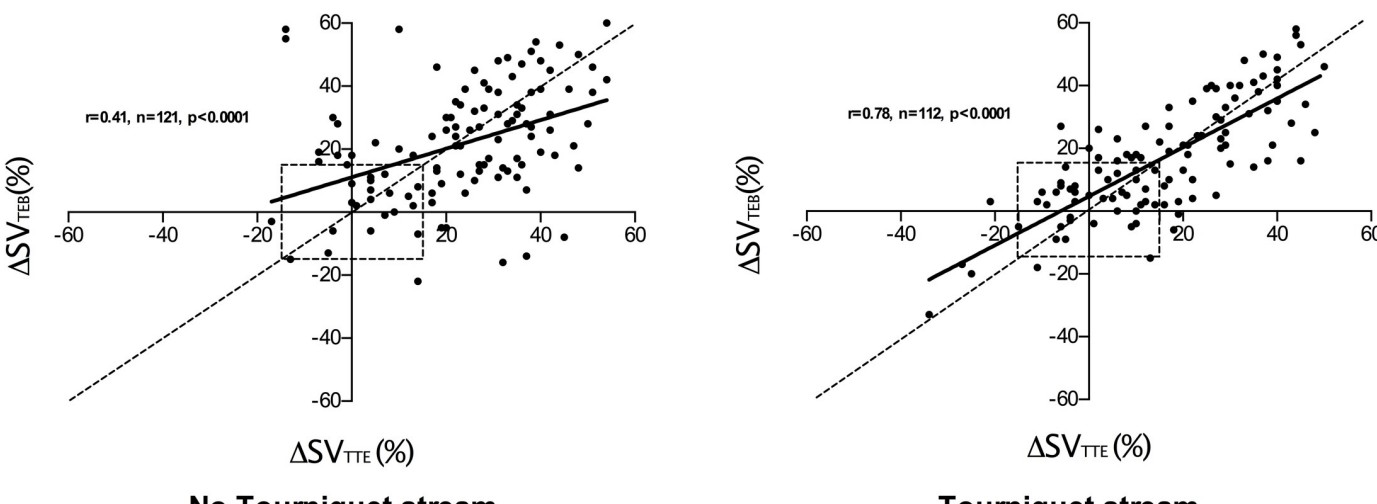

**Fig 4. 4 quadrant plot showing changes in stroke volume from baseline (ΔSV) recorded using trans thoracic bioimpedence (TBI) and trans-thoracic echocardiography (TTE).** 15% exclusion zone shown as dotted central square. Line of identity (y = x) shown as dotted line and regression line shown as solid line. r value indicates degree of correlation (Pearson's co-efficient).

Error grids comparing $\Delta SV_{TTE}$ and $\Delta SV_{TEB}$ are shown in Fig 6. In the non-tourniquet strand 121 matched values were plotted. The risk of harm as a result of a measurement error was classed as: none 75 (62%), mild 18 (15%), moderate 22 (18%) and severe 6 (5%). In the tourniquet strand 113 matched values were plotted. The risk of harm as a result of a measurement error was classed as: none 80 (71%), mild 17 (15%), moderate 16 (14%). No values indicated a risk of severe harm in this strand. The overall risk of severe harm when measurements for both streams were combined was 2.6%. 1 subject in the non-tourniquet strand had 3 measurement values indicative of severe harm, 1 subject had 2 such values and a third subject had a single value.

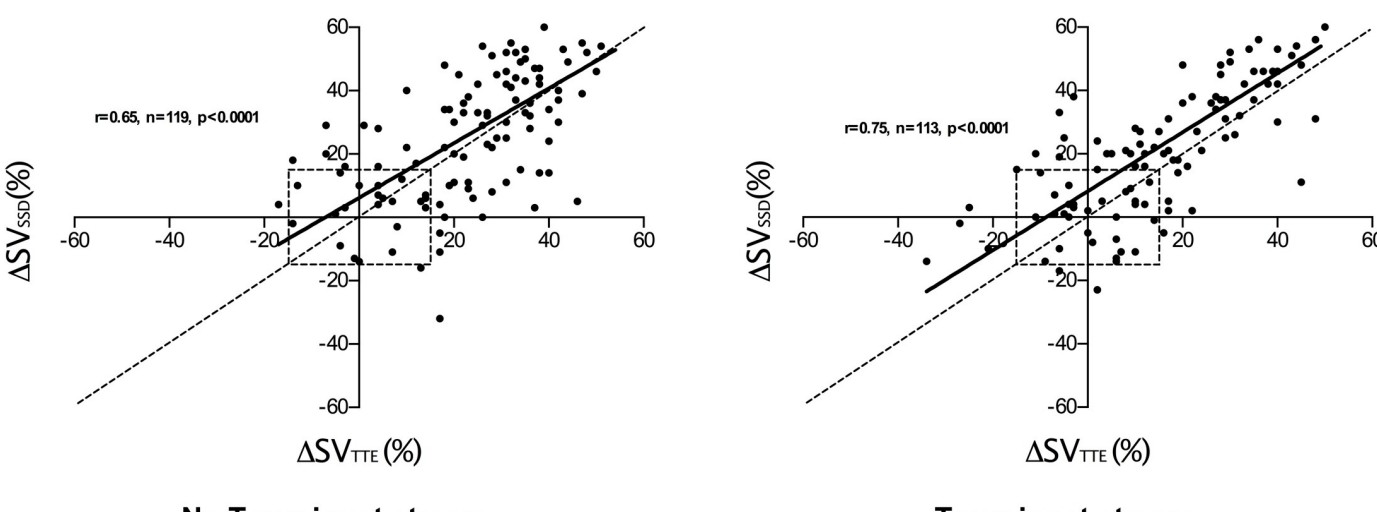

**Fig 5. 4 quadrant plot showing changes in stroke volume from baseline (ΔSV) recorded using supra-sternal Doppler (SSD) and trans-thoracic echocardiography (TTE).** 15% exclusion zone shown as dotted central square. Line of identity (y = x) shown as dotted line and regression line shown as solid line. r value indicates degree of correlation (Pearson's co-efficient).

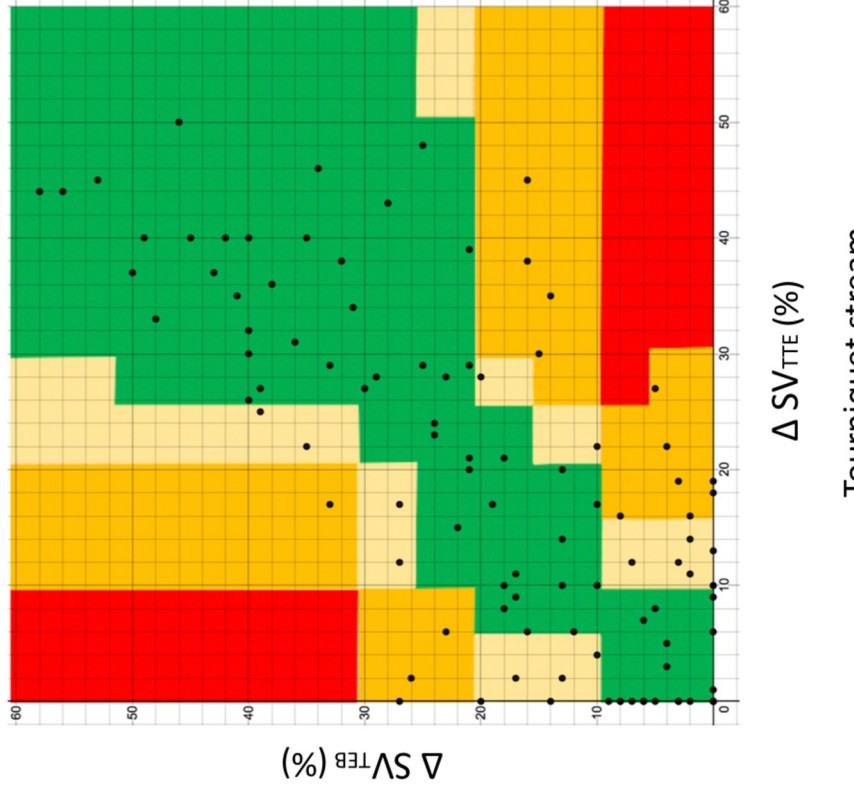

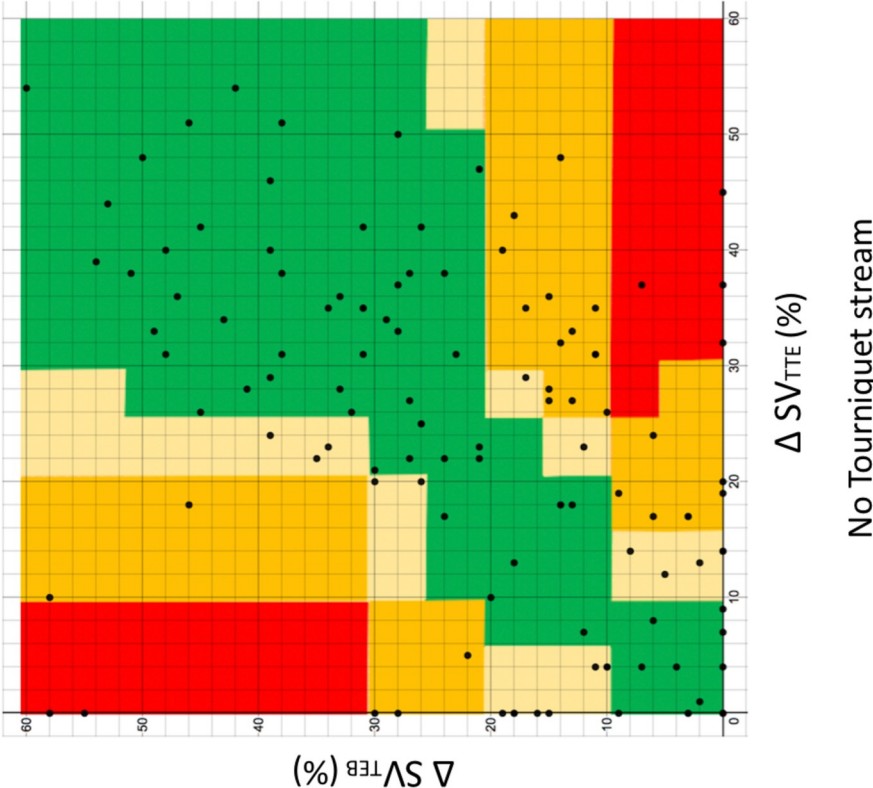

**Fig 6. Error grid showing changes in stroke volume from baseline (ΔSV) recorded using trans thoracic electrical bioimpedence (TEB) and trans-thoracic echocardiography (TTE).** Coloured zones represent the perceived degree of clinical harm resulting from a measurement error: Green = none, Yellow = mild, Orange = moderate, Red = severe.

Error grids comparing $\Delta SV_{TTE}$ and $\Delta SV_{SSD}$ are shown in Fig 7. In the non-tourniquet strand 121 matched values were plotted. The risk of harm as a result of a measurement error was classed as: none 79 (65%), mild 22 (18%), moderate 17 (14%) and severe 3 (2%). In the tourniquet strand 113 matched values were plotted. The risk of harm as a result of a measurement error was classed as: none 86 (76%), mild 14 (12%), moderate 11 (10%) and severe 2 (2%). The overall risk of severe harm when measurements for both streams were combined was 4.4%. 4 subjects, 2 in each strand, each had 1 measurement value indicative of severe harm.

## Discussion

The LBNP model achieved the aim of simulated hypovolaemia, producing a graded, sustained fall in stroke volume followed by a return to baseline when suction was discontinued. Most previous studies have used fixed degrees of applied LBNP in order to produce simulated hypovolaemia but this is limited by the inter-individual response and can produce a non-uniform reduction in stroke volume. Our approach was to target the degree of LBNP to a reduction in stroke volume recorded using finger plethysmography, a device that produces a continuous measure of SV. In this way we were able to produce a more standardised fall in SV across all of the study subjects. Although finger plethysmography was not a candidate device for comparison in the current study, principally because of its lack of utility in an urgent or emergency care setting, the SV values produced by this device correlated well to the reference standard.

The application of lower limb tourniquets were a feature of this study for two principal reasons. Firstly, we wished to assess whether the presence of a potentially increased afterload caused by the presence of the tourniquets produced divergent results for SV measurement in the two strands of the study. In fact, our results show very little discernible difference between

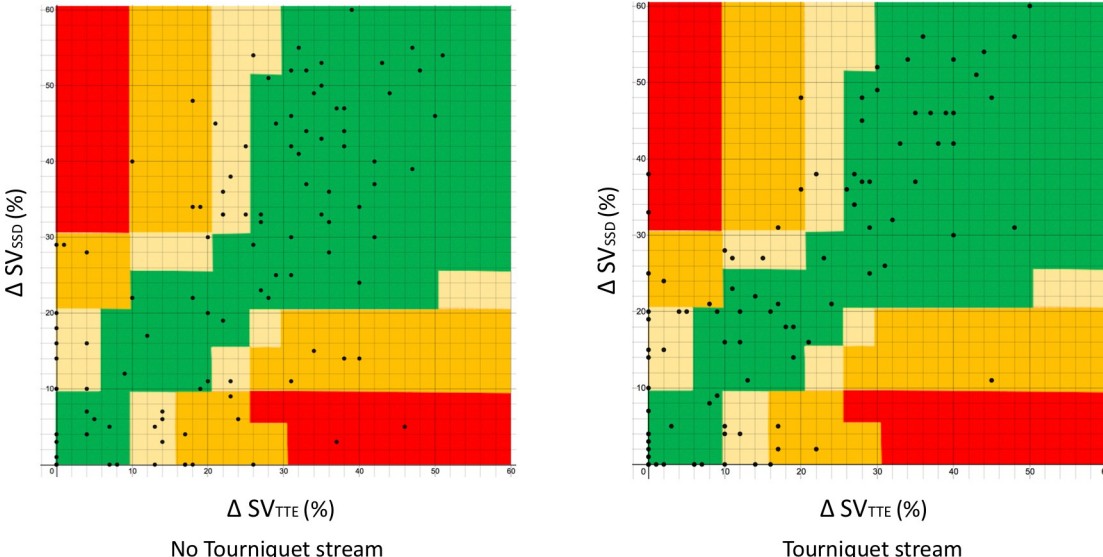

No Tourniquet stream    Tourniquet stream

**Fig 7. Error grid showing changes in stroke volume from baseline (ΔSV) recorded using suprasternal Doppler (SSD) and trans-thoracic echocardiography (TTE).** Coloured zones represent the perceived degree of clinical harm resulting from a measurement error: Green = none, Yellow = mild, Orange = moderate, Red = severe.

the two strands and it is likely that the addition of lower limb tourniquets did not produce a noticeable difference in the measured SV. Secondly we wanted, as far as possible, to reproduce the nociceptive response to trauma which has been shown to alter the haemodynamic response to haemorrhage [2].

Our results demonstrate that both non-invasive stroke volume monitors produced divergent values from the reference standard produced by echocardiography. Consensus opinion holds that a percentage error (PE) of 30% is the upper limit of acceptability in a clinical cardiac output monitoring device [15], however, both subject devices in the present study recorded errors well above this limit. In the case of the thoracic bioimpedance device there was also a very noticeable degree of bias which increased progressively at higher stroke volumes. Previous studies that have compared thoracic bio-impedence derived cardiac output to a reference standard have also shown that they are unreliable in producing accurate values for absolute measures of stroke volume. A meta-analysis examining the use of thoracic bioimpedance in nine studies showed only one instance where the PE value of < 30% with many reporting values well in excess of this threshold and up to 73% [17].

More recent studies conducted in pregnant women have reported that thoracic bioimpedance produces acceptable limits of agreement when compared to echocardiography but these studies do not induce any dynamic change in stroke volume in the study subjects, limiting the translatability of the results [18, 19].

The findings of the current study show that although supra-sternal Doppler was more accurate than thoracic electrical bio-imepedence it still showed a potentially unacceptable degree of error with a PE of 35% and 48%. These findings are in keeping with a meta-analysis of 6 studies that compared supra-sternal Doppler derived cardiac output to a reference standard and showed a mean PE of 42.7% [20].

Both devices performed better at assessing the change in SV rather than the absolute value, with concordance rates of around 90%, a value which has been suggested indicates an acceptable performance [15]. There was also a significant correlation between $\Delta SV_{TTE}$ and both $\Delta SV_{TEB/SSD}$, however, the observed correlation was not particularly strong, most notably in the bioimpedance non–tourniquet strand. It is worth commenting that the use of concordance rates in this setting may also give a falsely reassuring picture as to device performance. Although the majority of changes in values are in the same direction and hence the overall concordance rate is high there are a number of highly divergent readings. These discrepancies are more clearly identified by using an error grid methodology.

The use of error grid analysis was originally described as a way of assessing the clinical acceptability of point of care blood glucose testing [21]. Although error grid methodology has been used to assess clinically relevant differences in blood pressure measurements [16] and has been recommended as a potentially useful technique in cardiac output comparison studies [14], there are few published studies in this area. The major advantage of this methodology is in highlighting the clinical relevance of differences in measurements. In keeping with previous studies we used consensus expert opinion from experienced specialists to construct an error grid, with a clinical scenario written to reflect the key research question. Although the majority of measurements recorded by both candidate devices fell into the no risk zone a significant number produced errors that were perceived by the consensus panel to have the potential for clinical risk. In a very small minority of cases this harm was perceived to have been severe. The potential for both the commission (i.e. administration of potentially unnecessary blood products or fluid) and omission (failure to provide blood product or fluid resuscitation) of therapy was reflected in the results for both devices. The pattern of error in the supra-sternal Doppler group appeared to be different to that of the thoracic bioimpedance group; the former only having single isolated errors whilst the latter had clusters of multiple errors in each subject.

The results of the current study highlight the fundamental difficulty in using non-invasive stroke volume monitors in the initial management of patients with suspected blood loss. Firstly, whilst most clinicians would accept that a trend in a physiological variable is more useful as a gauge of the degree of blood loss or response to treatment, such a trend necessarily requires repeated measures. At the onset of a clinical scenario, clinicians must make an assessment based on the data to hand, and are therefore reliant on a one off measurement. It is this aspect that makes the measurement of systolic blood pressure such a seductive target as population baseline values are widely understood, even by individuals with limited clinical training and experience. However, as discussed and demonstrated in the current study, blood pressure is often a poor measure of the degree of blood loss. What is needed is a quick and precise way of ascertaining stroke volume and hence blood flow and then indexing that value to the size of the patient in order to provide a more accurate measure of intra vascular volume status. The results of the current study do not demonstrate that either of the candidate devices is capable of this. Although, both devices were better at detecting trends in stroke volume changes, with apparently acceptable concordance rates, the use of error grid analysis clearly highlights the presence of a small number of highly aberrant results, produced by both devices, which if acted upon could have the potential to produce clinical harm.

Our study has several limitations. Firstly although the LBNP technique is well described and efficacious in producing a change in stroke volume over a clinically relevant range by reducing venous return, our study is experimental and not clinical which may limit the translation of these data into clinical practice. However, controlled experiments of flow monitoring in actual traumatic haemorrhage are highly unlikely to be achievable. An additional limitation is the use of healthy subjects with no cardiovascular disease and the absence of cardiovascular changes induced by vasoactive drugs and large infusions of fluid. Finally, whilst the thoracic bioimpedance device is essentially non operator dependent both supra-sternal Doppler and our chosen reference standard for stroke volume assessment rely on a user acquiring an optimised Doppler signal. We mitigated this limitation by using a single user for both techniques across the entire study and ensuring that both users were trained and experienced in the technique.

In conclusion we found that two flow monitoring devices, based on different physical principles, both had poor accuracy in measuring absolute stroke volume when compared to a reference standard. Both devices were better able to detect trend changes in stroke volume in a simulated hypovolaemia model but in a small minority of cases produced measurement errors that had the potential to produce significant clinical harm. If such devices are used for the early detection of hypovolaemia following haemorrhage, the values produced should be interpreted with caution and not used to determine therapy in isolation.

## Supporting information

**S1 File. Error grid questionnaire.** Background material and information sent to respondents in order to provide material to construct error grids.
(DOCX)

**S2 File. Error grid questionnaire returns.** Error grid questionnaire results from 15 respondents. Respondents first provided information on what fall in stroke volume from baseline would either require no action (Code A), possible action (Code B) or essential action (Code C)–these results are shown on the ACTUAL column of each table. Respondents were then asked to quantify the harm from a divergent measurement using the same range of stroke volumes–shown as DEVICE in the table. Harm was quantified numerically as None (0), Mild (2), Moderate (5) or Severe (10). Cumulative results are shown in a single table in which a range of

actual versus measured falls in stroke volume are shown along with the perceived degree of harm from measurement error. The range of harm is from 0 (0 respondents thought that harm could occur) to 150 (all respondents thought that severe harm was likely).
(DOCX)

**S3 File. Error grid results.** Excel spreadsheet where each cell is a comparator between actual fall in stroke volume and measured fall in stroke volume. The range of harm from 0–150 has been recalculated as a percentage value and colour coded using the colour scales function.
(XLSX)

**S1 Fig. Error grid polygons.** Smoothed polygon created from the Excel spreadsheet presented in File S3. Colours indicate degree of harm Red (Severe), Orange (Moderate), Yellow (Mild), Green (None).
(PNG)

# Author Contributions

**Conceptualization:** Sam D. Hutchings, Emrys Kirkman.

**Data curation:** Sam D. Hutchings, Emrys Kirkman.

**Formal analysis:** Sam D. Hutchings, Jim Watchorn, Sarah Watts, Emrys Kirkman.

**Funding acquisition:** Sam D. Hutchings, Sarah Watts, Emrys Kirkman.

**Investigation:** Sam D. Hutchings, Jim Watchorn, Rory McDonald, Su Jeffreys, Mark Bates, Emrys Kirkman.

**Methodology:** Sam D. Hutchings, Emrys Kirkman.

**Project administration:** Sam D. Hutchings, Su Jeffreys, Emrys Kirkman.

**Supervision:** Sam D. Hutchings.

**Validation:** Sam D. Hutchings, Emrys Kirkman.

**Writing – original draft:** Sam D. Hutchings.

**Writing – review & editing:** Sam D. Hutchings, Jim Watchorn, Rory McDonald, Mark Bates, Emrys Kirkman.

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
