## [Decision Letter · Decision Letter 0]

6 Sep 2021

PONE-D-21-19494Detection of hypovolaemia in a simulated healthy volunteer model of traumatic haemorrhage; a comparison of two non-invasive monitoring devices using error grid analysis alongside traditional measures of agreementPLOS ONE

Dear Dr. Hutchings,

Thank you for submitting your manuscript to PLOS ONE. After careful consideration, we feel that it has merit but does not fully meet PLOS ONE’s publication criteria as it currently stands. Therefore, we invite you to submit a revised version of the manuscript that addresses the points raised during the review process.

ACADEMIC EDITOR: I appreciated the importance of your work. The reviewers were positive about your manuscript, but one of them requested revisions. Please address all the comments pointed out by the reviewer. You will find that the comments help improve the manuscript.

We look forward to receiving your revised manuscript.

Kind regards,

Kenta Matsumura

Academic Editor

PLOS ONE

Journal Requirements:

2. Please include additional information regarding the survey or questionnaire used in the study and ensure that you have provided sufficient details that others could replicate the analyses. For instance, if you developed a questionnaire as part of this study and it is not under a copyright more restrictive than CC-BY, please include a copy, in both the original language and English, as Supporting Information. If the original language is written in non-Latin characters, for example Amharic, Chinese, or Korean, please use a file format that ensures these characters are visible.

3. Please state whether you validated the questionnaire prior to testing on study participants. Please provide details regarding the validation group within the methods section.

Reviewers' comments:

Reviewer's Responses to Questions

**Comments to the Author**

1. Is the manuscript technically sound, and do the data support the conclusions?

Reviewer #1: Yes

Reviewer #2: No

2. Has the statistical analysis been performed appropriately and rigorously? 

Reviewer #1: I Don't Know

Reviewer #2: No

3. Have the authors made all data underlying the findings in their manuscript fully available?

Reviewer #1: Yes

Reviewer #2: Yes

4. Is the manuscript presented in an intelligible fashion and written in standard English?

Reviewer #1: Yes

Reviewer #2: Yes

5. Review Comments to the Author

Reviewer #1: In this report, hypovolaemia which is simulated by lower body negative pressure (LBNP) method is studied by detecting a stroke volume (SV). The SV is measured by two devices, trance-thoracic electrical bioimpedance (TEB) and supra-sternal Doppler (SSD), and is compared between two devices against a reference standard measured with using trans-thoracic echocardiography (TTE).

Results show that these two devices have poor accuracy in measuring absolute stroke volume when compared to a reference standard. Authors describe that both devices are more useful for detecting trend changes in the SV than the absolute value; the obtained value with two device should be interpreted with caution.

Measurement reliability is most important for medical workers using these device, and is essentially judged only by accumulation of a large amount of data. Hence this paper is an important contribution and I recommend that it be accepted for publication.

Reviewer #2: I found the topic interesting, and the study appears to be well executed. However, what the authors did in this study was not detection of hypovolaemia but examination of agreement of SV of different devices using LBNP technique. Although data itself is potentially useful, the authors need to change the title, introduction, and discussion to reflect what they did in this study.

Introduction:

Please summarize hemodynamic changes induced by LBNP referring to previous studies (e.g., Guo, J Appl Physiol,100: 1785–1792, 2006. Levine, Circulation 90: 298–306, 1994).

Based on the above, the authors should mention the weak points of former works (identification of the gaps) and describe the current investigation's novelties to justify the paper deserves to be published in PLOS ONE.

Methods:

"21 subjects": Was the alpha level set to 0.05?

Please define "SSD"

Body mass index —> "k"g/m2

Please add references regarding the Bland-Altman plot.

"In order to…error grid methodology).": Is this necessary? If so, please summarize the results in the Results section.

Results:

Please add the results of a series of repeated ANOVAs, such as F (20, 400) = 34.56, partial eta squared = 0.63, p = 0.0024, by main effect and interaction, to the main text.

Discussion:

The authors will need to revise this section according to the above changes.

6. PLOS authors have the option to publish the peer review history of their article (what does this mean?). If published, this will include your full peer review and any attached files.

Reviewer #1: No

Reviewer #2: No

---

## [Author Response · Author response to Decision Letter 0]

29 Oct 2021

Journal Requirements 

1. The manuscript has been reformatted in accordance with the journal requirements 

2. A copy of the questionnaire is included in the supporting information (S1 File). 

3. The questionnaire was not validated prior to the study 

4. ORCID ID included in editorial manager 

5. Supporting information citations re formatted within manuscript 

Responses to Reviewer 2 

Reviewer #2: I found the topic interesting, and the study appears to be well executed. However, what the authors did in this study was not detection of hypovolaemia but examination of agreement of SV of different devices using LBNP technique. Although data itself is potentially useful, the authors need to change the title, introduction, and discussion to reflect what they did in this study.

The title has been amended to include the phrase assessment of stroke volume. 

Introduction:

Please summarize hemodynamic changes induced by LBNP referring to previous studies (e.g., Guo, J Appl Physiol,100: 1785–1792, 2006. Levine, Circulation 90: 298–306, 1994).

Based on the above, the authors should mention the weak points of former works (identification of the gaps) and describe the current investigation's novelties to justify the paper deserves to be published in PLOS ONE.

Section added to the introduction providing more granularity on the LBNP technique – suggested references cited 

Methods:

"21 subjects": Was the alpha level set to 0.05?

Yes alpha 0.05 – added to manuscript 

Please define "SSD"

“Supra sternal Doppler” – clarification added 

Body mass index —> "k"g/m2

Corrected 

Please add references regarding the Bland-Altman plot.

Actioned

"In order to…error grid methodology).": Is this necessary? If so, please summarize the results in the Results section.

I’m afraid I don’t understand this comment. Error grid methodology is a core part of the paper and the results are clearly presented. 

Results:

Please add the results of a series of repeated ANOVAs, such as F (20, 400) = 34.56, partial eta squared = 0.63, p = 0.0024, by main effect and interaction, to the main text.

The requested data has been added in Table 1

---

## [Decision Letter · Decision Letter 1]

1 Dec 2021

PONE-D-21-19494R1Quantification of stroke volume in a simulated healthy volunteer model of traumatic haemorrhage; a comparison of two non-invasive monitoring devices using error grid analysis alongside traditional measures of agreementPLOS ONE

Dear Dr. Hutchings,

Thank you for submitting your manuscript to PLOS ONE. After careful consideration, we feel that it has merit but does not fully meet PLOS ONE’s publication criteria as it currently stands. Therefore, we invite you to submit a revised version of the manuscript that addresses the points raised during the review process.

The revised version of your study satisfied most of the concerns raised by the reviewers. However, the reviewer raised additional comments that would strengthen your analysis. Thus, I would like to invite you to submit a re-revised version of the paper for consideration again. If your response is comprehensive, I will likely review the changes myself rather than send the paper out for re-review.

We look forward to receiving your revised manuscript.

Kind regards,

Kenta Matsumura

Academic Editor

PLOS ONE

Reviewers' comments:

Reviewer's Responses to Questions

**Comments to the Author**

Reviewer #2: All comments have been addressed

2. Is the manuscript technically sound, and do the data support the conclusions?

Reviewer #2: Partly

3. Has the statistical analysis been performed appropriately and rigorously? 

Reviewer #2: Yes

4. Have the authors made all data underlying the findings in their manuscript fully available?

Reviewer #2: Yes

5. Is the manuscript presented in an intelligible fashion and written in standard English?

Reviewer #2: Yes

6. Review Comments to the Author

Reviewer #2: Thank you for your revision. I am satisfied with most responses. But, I suggest that the error grid protocol be incorporated into the main text, not as supplementary material, and mention its novelty in the introduction. These would strengthen your study.

7. PLOS authors have the option to publish the peer review history of their article (what does this mean?). If published, this will include your full peer review and any attached files.

Reviewer #2: No

---

## [Author Response · Author response to Decision Letter 1]

2 Dec 2021

We have incorporated the error grid methodology in the main methods section of the manuscript and added a paragraph to the introduction expanding on the role of error grid methodology in device comparison studies.

---

## [Editor Report · Decision Letter 2]

6 Dec 2021

Quantification of stroke volume in a simulated healthy volunteer model of traumatic haemorrhage; a comparison of two non-invasive monitoring devices using error grid analysis alongside traditional measures of agreement

PONE-D-21-19494R2

Dear Dr. Hutchings,

We’re pleased to inform you that your manuscript has been judged scientifically suitable for publication and will be formally accepted for publication once it meets all outstanding technical requirements.

Kind regards,

Kenta Matsumura

Academic Editor

PLOS ONE
---

## [Editor Report · Acceptance letter]

10 Dec 2021

PONE-D-21-19494R2 

Quantification of stroke volume in a simulated healthy volunteer model of traumatic haemorrhage; a comparison of two non-invasive monitoring devices using error grid analysis alongside traditional measures of agreement 

Dear Dr. Hutchings:

I'm pleased to inform you that your manuscript has been deemed suitable for publication in PLOS ONE. Congratulations! Your manuscript is now with our production department. 

Kind regards, 

on behalf of

Dr. Kenta Matsumura 

Academic Editor

PLOS ONE